



# The Entrainment of Air from Rainy Surface Regions and its Implications for Bioaerosol Transport in Three Deep Convective Storm Morphologies

Charles M. Davis[1], Susan C. van den Heever[1], Leah D. Grant[1], Sonia M. Kreidenweis[1], Claudia Mignani[1,2], Russell J. Perkins[1], and Elizabeth A. Stone[3]

[1]Department of Atmospheric Science, Colorado State University, Fort Collins, CO, 80521, USA
[2]Water and Soil Resource Research, Institute of Geography, University of Augsburg, Augsburg, 86159, Germany
[3]Department of Chemistry, University of Iowa, Iowa City, IA, 52242, USA

*Correspondence to*: Charles M. Davis (cmdavis4@colostate.edu)

**Abstract.** The rain produced by thunderstorms has been observed to coincide spatially and temporally with enhanced near-surface concentrations of warm-temperature ice nucleating particles (INPs) of biological origin. However, the air in rainy regions is evaporatively cooled and negatively buoyant, and so it is unclear if it is entrained into its parent storms. Despite bioaerosols being highly ice-nucleation active, the microphysical influence that rain-aerosolized bioaerosols exert on storm processes is therefore not well-understood. We use the RAMS cloud-resolving model to simulate high-resolution archetypal representations of three deep convective storm morphologies: isolated deep convection, a squall line, and a supercell. We measure the degree of entrainment of rainy and non-rainy surface air into its parent storm using passive tracers, as well as calculating measures of each storm's characteristics that influence the timing and degree of this entrainment. We find different degrees of entrainment between storm morphologies and between rainy and non-rainy surface air, with the squall line and supercell entraining significantly more rainy air than the isolated convective storm for all but the lightest rain. These differences owe to variation between the storms in their degrees of entrainment of surface air, their proportions of entrained surface air that originate in rainy regions, and their amount of rain produced per updraft mass. This study finds a specific and previously unrecognized source of air potentially containing highly ice-active aerosols which is entrained to varying degrees in different convective storm morphologies, and which is likely to exert different microphysical impacts on each type of storm.

## 1 Introduction

The origins of the air and particles that are entrained into the updrafts of convective storms has been a longstanding area of research. The advent of high-resolution cloud-resolving models has greatly facilitated the investigation of questions related to the origins of air in convection and other settings. For example, Fierro et al. (2009) used such an approach to investigate and refine the hot tower hypothesis of Riehl and Malkus (1958). They confirmed the proposed transport of tropical surface





air to the upper troposphere by deep convective towers but found significant dilution of these purportedly undiluted towers. They also demonstrated that more than half of the parcels initialized in the boundary layer ahead of a mature maritime squall line are lofted to at least 10 km mean sea level, with the primary entrainment pathway being gust front lofting into the storm's updraft. Applying a similar Lagrangian analysis of parcel trajectories in a simulation of a tropical deep convective

squall line, McGee and van den Heever (2014) determined that the air in the strongest regions of the storm's convective updrafts originates near the surface, but that most of the air that is lifted to altitudes of 10 km or more originates from air layers that are more than 2 km above the surface.

        Eulerian approaches, like that of this study, have also been employed to understand the origins of aerosols transported by storms. Seigel and van den Heever (2012) analyzed the pathways by which dust is entrained into a simulated

supercell under several different dust emission scenarios. They found that significant concentrations of dust are entrained into the storm's updraft via gust front lofting when the dust originated ahead of the cold pool (i.e. in a dusty background environmental state), but that minimal dust is entrained into the storm's updraft when its only emission mechanism is lofting by the surface winds associated with the storm's cold pool. This is despite the fact that large concentrations of dust are lofted into the cold pool itself; the dynamics of the storm are such that the dust in the cold pool can only be entrained into the

updraft when it is first detrained from a small region at the head of the cold pool by turbulent mixing, which only occurs in small quantities. Grant et al. (2018) modelled tropical mesoscale convective systems (MCSs) and, using a passive tracer released at the surface within the storms' cold pools, found that ~17-21% of this tracer was lifted to at least 3 km above ground level (AGL). The difference between this finding and that of Seigel and van den Heever (2012) surely arises in large part from the different storm morphologies considered and tropical versus midlatitude environments. However, another

difference is in the use of tracers, which are subject only to advection and diffusion and therefore do not represent aerosols directly, versus true dust aerosols. As aerosols are subject to other processes as well (activation, settling, deposition, radiative effects), we might well expect to see differences in the degree of entrainment resulting from these differing representations in these studies. We further discuss this point as it pertains to the present study in Sect. 2.2.

        Other research has specifically tried to ascertain the vertical level at which microphysically active aerosols in deep

convective storms originate. Existing studies have found that midtropospheric aerosols are of critical importance for convective storm microphysics: they are the predominant formation site of anvil cirrus ice crystals (Fridlind et al., 2004), they are the primary source of aerosols in the mixed phase region of an idealized MCS (Lebo, 2014), and they enhance mixed-phase microphysical processes in simulated MCSs (Lebo, 2014; Marinescu et al., 2017). Existing work on the importance of low-level aerosols has found more mixed results, however. In the foregoing studies, for instance, Fridlind et

al. (2004) found that midtropospheric aerosols are of the greatest importance for anvil cirrus microphysics and Lebo (2014) found that low-level aerosols have minimal impact on the strength of their simulated MCS, while Marinescu et al. (2017) found that low-level aerosols enhance precipitation in the cold pools of their simulated MCSs, which in turn leads to more evaporative cooling and faster propagation speeds. Similarly, van den Heever et al. (2006) found significant enhancements to updraft strength and mixed-phase processes from the introduction of low-level aerosols in a simulation of an observed





subtropical convective storm. The influence of low-level aerosols on the microphysics of convective storms is, therefore, a topic where many open questions remain.

Other research has analyzed the impact of aerosol deposition processes on the convective transport of aerosols. Tulet et al. (2010) used a mesoscale model to simulate an observed MCS over Niger and found that MCSs can transport dust aerosols to the tropopause in concentrations up to 6 particles cm$^{-3}$. They also noted that the inclusion of wet dust scavenging

processes in their model reduced dust concentrations in convective cores from 50 µg m$^{-3}$ to less than 1 µg m$^{-3}$. Herbener et al. (2016) similarly demonstrated that a tropical cyclone's convective updraft transports dust to the upper troposphere, but that 75x as much dust is returned to the surface via wet and dry deposition than is transported to the upper troposphere. The implications of these studies for our results, which use inert tracers that are not subject to these processes, are discussed further in Sect. 4.

No work to date has investigated the entrainment pathways or ultimate fate of biological aerosol particles (bioaerosols) specifically. The transport of bioaerosols in deep convective storms is likely to be consequential for several reasons. First, bioaerosols nucleate ice at the warmest temperatures of any currently known class of aerosol particles (Després et al., 2012; Hoose and Möhler, 2012; Joly et al., 2014; Morris et al., 2014). Although this fact on its own would point toward the potential for these particles to microphysically influence the development of storms, existing work on

whether bioaerosols influence storms in practice has found conflicting results depending on the spatial and temporal scale considered, as well as on the specifics of the bioaerosol microphysical parameterization employed (Diehl et al., 2002; Hoose et al., 2010; Patade et al., 2022; Wozniak et al., 2018). Second, bioaerosols are known to have unique emission mechanisms associated with the impact of rain on the earth's surface. These mechanisms include the ejection of very small droplets containing bacteria from the soil upon a raindrop's impact onto the surface (Joung et al., 2017), mechanical aerosolization of

plant detritus and fungal spores when raindrops strike plant leaves (Jones and Harrison, 2004; Mignani et al., 2025; Tobo et al., 2013), and the rupture of pollen grains due to the high humidity associated with rain events (Hughes et al., 2020; Suphioglu et al., 1992; Taylor et al., 2002). While several recent modelling studies have looked at the impacts of bioaerosols on storms (Subba et al., 2023; Werchner et al., 2022; Zhang et al., 2024), no previous study has either examined the storm-scale entrainment pathways of these particles or implemented a rain-induced emission mechanism. Additionally, the

phenomenon of "thunderstorm asthma" in which previously healthy people experience respiratory distress following a storm is thought to be a result of high exposures to respirable bioaerosols in near-surface aerosol before and after storms (D'Amato et al., 2008, 2016; Newson et al., 1997; Packe and Ayres, 1985; Pulimood et al., 2007). Understanding the transport of bioaerosols in storms thus has implications for public health.

As the existing evidence for bioaerosol influence on convective storms is conflicting, and as questions of the

transport of bioaerosols within such storms have not been fully addressed, it is important to understand their transport both for its own sake and for its contribution to understanding bioaerosol-cloud interactions. This study seeks to extend our understanding of entrainment and transport dynamics of bioaerosols in deep convective storms by analyzing these processes in high-resolution simulations. Our idealized simulations produce archetypal examples of each storm morphology that we





consider so that our findings are as applicable as possible to other instances of these storm morphologies, whether in
simulations or in nature.

## 2 Methods

### 2.1 Idealized model setup

To address our research goals, we conduct a suite of idealized simulations using the Regional Atmospheric Modeling System
(RAMS) version 6.3.04 (Cotton et al., 2003; van den Heever et al., 2022; Saleeby and van den Heever, 2013). We simulate
well-established representations of three deep convective storm morphologies: an isolated convective storm (Kingsmill and
Wakimoto, 1991; Moroda et al., 2021; Schlesinger, 1978; Szoke and Zipser, 1986; Wilhelmson, 1974; Yang et al., 2016), a
squall line (Bluestein and Jain, 1985; Browning, 1977; Byers and Braham, 1949; Maddox, 1980; Newton, 1950; Rotunno et
al., 1988; Seigel and van den Heever, 2013; Smull and Houze, 1987; Weisman and Klemp, 1984; Weisman and Rotunno,
2004), and a supercell (Browning, 1964; Davies-Jones, 1984; Fujita and Grandoso, 1968; Grant and van den Heever, 2014;
Lilly, 1982, 1983; Newton and Katz, 1958; Schlesinger, 1978). We use the term "isolated convective storm" throughout this
work to refer to a common weakly-sheared "air mass" convective storm, and not to refer to the supercell although it is
technically an isolated convective storm. We will also at times refer to the squall line and supercell collectively as the "more
organized" storms. These three storm morphologies comprise the majority of summertime convective storms in the
continental midlatitudes (Cotton et al., 2010), and thus are likely to be most responsible for rain-induced aerosolization of
biological particles in these regions.

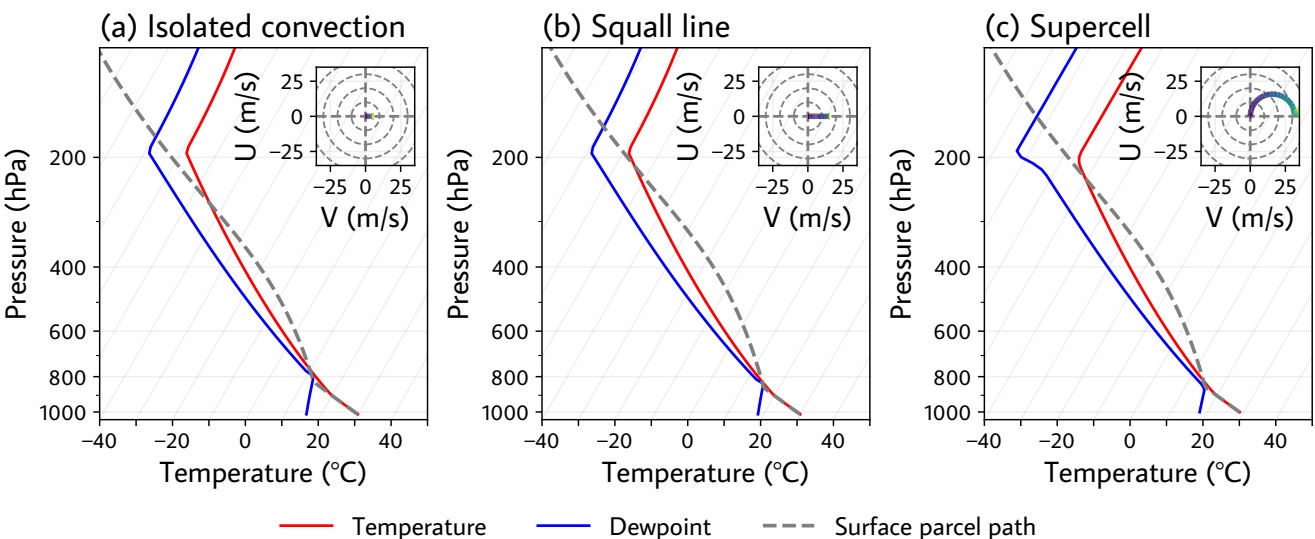

**Figure 1 Skew-T log-P diagrams of the initial environments of each storm morphology. All simulations were initialized with a
Weisman-Klemp (1984) sounding. The low-level moisture and shear profile were varied between storms, as detailed in Table 1.
The hodographs show the winds from 0 km – 8 km AGL to more clearly demonstrate the shear profiles.**




| Model Parameter | Description | | |
|---|---|---|---|
| | **Isolated deep convection** | **Squall line** | **Supercell** |
| **Model** | RAMS v6.3.04 (Cotton et al., 2003; van den Heever et al., 2022; Saleeby and van den Heever, 2013) | | |
| **Timestep** | 1 s | | |
| **Output frequency** | 5 minutes | | |
| **Microphysics** | Full bin-emulating bulk microphysics (Meyers et al. 1997) with 2 liquid (rain and cloud water) and 5 ice hydrometeor species (Saleeby and Cotton 2004), ice nucleation parameterization described in DeMott et al. (2010) | | |
| **Radiation** | Off | | |
| **Horizontal grid spacing** | 500 m | | |
| **Vertical grid spacing** | Starts at 25m, stretched with a stretching ratio of 1.1 up to a max spacing of 250m and max height of 22.8km | | |
| **Turbulence** | Smagorinsky (Smagorinsky 1963) with stability modifications by Lilly (1962) and Hill (1974) | | |
| **Top boundary** | Rayleigh dampening layer applied in the upper 2km with a dissipation timescale of 60s | | |
| **Horizontal boundaries** | x: radiative<br>y: radiative | x: radiative<br>y: cyclic | x: radiative<br>y: radiative |
| **Convective initiation** | 2 K warm bubble, 80 km x 80 km x 1.2 km, amplitude Gaussian with distance from center | -4 K cold bubble, uniform in y, 20 km x 3.8 km in x and z, respectively; Gaussian amplitude in x and z | 2 K warm bubble, 17 km x 17 km x 3.0 km, amplitude Gaussian with distance from center |
| **Domain size** | 375km x 220km | 475km x 375km | 425km x 300km |
| **Weisman-Klemp (1984) initial profile** | 5 m s$^{-1}$ shear over 0 km to 3 km AGL<br>Straight hodograph<br>11 g kg$^{-1}$ max vapor mixing ratio | 15 m s$^{-1}$ shear over 0 km to 3 km AGL<br>Straight hodograph<br>13 g kg$^{-1}$ max vapor mixing ratio | 34 m s$^{-1}$ shear over 0 km to 7.5 km AGL<br>Curved hodograph<br>13 g kg$^{-1}$ max vapor mixing ratio |
| **Initial aerosol profile** | Sulfates, log-normally distributed with median radius of 75 nm; concentration decreases exponentially with height, from 500 mg$^{-1}$ at the surface with e-folding height of 7 km | | |

**Table 1: The RAMS settings used in simulations conducted for this research.**



For all three storm types, we use a horizontal grid spacing of 500 m and a stretched vertical grid with vertical grid spacing ranging from 25 m at the surface to 250 m at higher levels. This fine vertical grid spacing at the surface is needed to
accurately represent low-level storm processes that are important to transport, in particular the dynamics of the cold pool and the low-level convective updraft. All three storms are initialized via warm- or cold-bubble forcings in an initial Weisman and Klemp (1984) environmental profile applied horizontally uniformly over the domain (Figure 1). The initial environmental soundings vary between the simulated storms only in the vertical shear profile and the low-level moisture, as described in Table 1, and follow the general approach initially established in Weisman and Klemp (1984). We do not include any
topography in the model domain's terrain.

Similarly to previous research, the isolated convective storm and the supercell are initialized via Gaussian warm bubbles (Grant and van den Heever, 2014; Seigel and van den Heever, 2012; Weisman and Klemp, 1982). The squall line is initialized via a cold bubble (Bryan and Morrison, 2012; Seigel and van den Heever, 2013; Weisman et al., 1997), the amplitude of which is zonally Gaussian about its center and meridionally uniform (i.e. the bubble is "infinitely" long
meridionally) (Mulholland et al., 2021; Seigel and van den Heever, 2013). We introduce random perturbations to the surface potential temperature with a maximum amplitude of 0.1 K throughout the domain of the squall line to break its meridional symmetry, and throughout the domains of the other storms for consistency. All lateral boundary conditions are radiative (Klemp and Wilhelmson, 1978), except for the meridional boundary of the squall line domain which is periodic. We employ a two-moment bulk microphysics scheme (Meyers et al., 1997). Radiation is disabled and free-slip lower boundary
conditions are used for all three simulations. Further model setup details are listed in Table 1.

**2.2 Passive tracer setup**

We utilize passive tracer quantities to measure the entrainment of surface air and its subsequent distribution within the storms. The tracer quantities are subject only to advection and diffusion and are implemented as additional variables in the numerical model. These tracer quantities can mix between air masses and move within the domain, but do not affect the
motion of the air or any other physical aspect of the simulation. We utilize separate tracer "species" which do not interact with each other to measure multiple origins and associated transport pathways in the same simulation.

Tracers are emitted into the model domain via two different source mechanisms, which we use to measure different air origins and transport pathways. We refer to the first of these source mechanisms as *fixed-source* (FS) emission. Fixed-source emission adds tracer to all horizontal gridpoints in the model domain at the lowest vertical level above the ground
(~12 m), at a constant rate and at each timestep. We utilize tracers emitted in this way (which we will refer to as fixed-source tracers) to quantify the amount of surface air that has been entrained into the storms. The tracer species emitted via this mechanism is changed after every five minutes of simulation time. As an example, tracer species 1 (TS1) is emitted in the manner just described from t=0 minutes to t=5 minutes. At t=5 minutes, this tracer is no longer emitted, and the amount emitted up to that time is all of the TS1 that will ever be emitted. Another tracer species, TS2, is then emitted beginning at
t=5 minutes. The per-timestep and per-gridpoint rate at which TS2 is emitted is identical to that of TS1. TS2 is emitted until





t=10 minutes, after which no more of it is ever emitted, and TS3 is then emitted at the same rate from t=10 minutes to t=15 minutes. This pattern continues for the 3 hours of simulation time for all three storm morphologies (Figure 2(a)). This approach has two desirable properties. First, the total fixed-source tracer emitted into the domain (i.e. the sum of the amount emitted across the fixed-source tracer species) is linear in time, which facilitates comparison of fixed-source tracer

concentrations across times and between storms. Second, changing the emitted species every five minutes allows us to identify the time at which a parcel of air was in contact with the surface by comparing concentrations between individual tracer species.

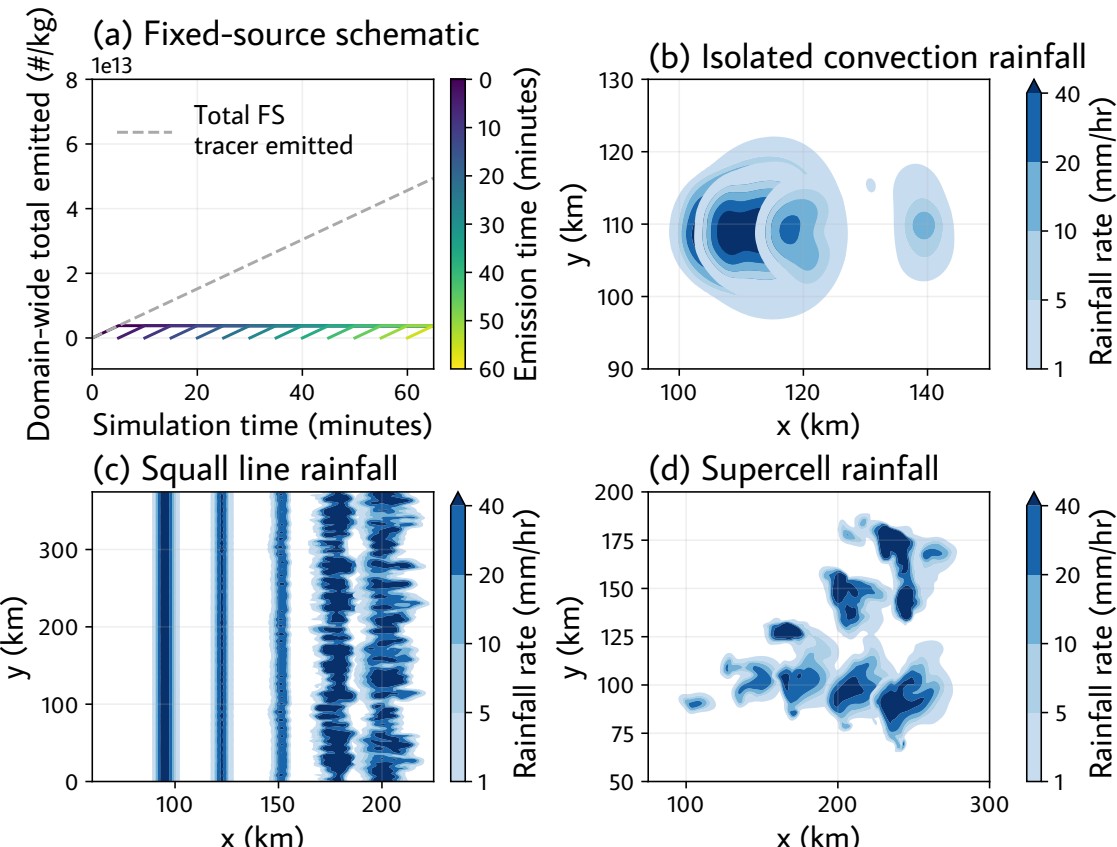

**Figure 2 Schematics of the fixed-source and rain-sourced tracer emission setups. Panel (a) shows the total amount of fixed-source**
**tracer emitted in the whole domain versus time, by individual tracer species and in total. Panels (b-d) show plan view contours of**
**the surface rain rate at 30-minute intervals in each storm. The rainfall rate contours shown correspond to the five rainfall rate**
**thresholds for emission of each of the five rain-sourced tracer species (as described in the text), and therefore to regions where**
**each tracer species is being emitted at that point in time. Note that the isolated convective storm has a smaller bulk velocity than**
**either of the other storms, and thus its contours at different points in time overlap each other to a greater degree.**

We refer to the other tracer emission mechanism as *rain-sourced emission* (and tracers emitted in this way as *rain-sourced tracers*). Rain-sourced tracer is emitted at each timestep in which the surface rain rate in a gridpoint exceeds a specified threshold value. Rain-sourced tracer emission is either "on" or "off" in that the emission rate does not depend on





the amount by which the rainfall threshold is exceeded. The per-timestep emission rate of the rain-sourced tracers is equal to that of the fixed-source tracers to facilitate comparison between the two tracer categories. This emission rate is 25,000 # kg$^{-1}$

s$^{-1}$ in each gridpoint. All tracer concentrations presented in this work are either compared to another tracer concentration or are normalized by the total amount of tracer emitted, and thus this choice of emission rate does not influence our findings and has no physical meaning. Five rain-sourced tracer species are utilized, each of which is emitted above a different instantaneous surface rain rate threshold of 1 mm hr$^{-1}$, 5 mm hr$^{-1}$, 10 mm hr$^{-1}$, 20 mm hr$^{-1}$, or 40 mm hr$^{-1}$ (Figure 2 (b–d)). These separately tracked tracer species allow us to differentiate the relative entrainment of air from regions of light vs

intense rainfall, as well as to compare entrainment from rainy regions vs non-rainy regions.

We emphasize that tracers are subject only to advection and diffusion, while aerosol particles would be subject to deposition and other aerosol-specific processes. These tracer quantities are therefore not intended to emulate actual aerosol particles, nor the aerosolization of such particles, but rather to track the location of *air* throughout these storm types that would contain biological particles aerosolized by rain.

**2.3 Definition of supercell right-mover**

Because we impose a veering environmental wind profile in the simulation of the supercell, the right mover is favored after the initial splitting of the storm. As we would expect, the right mover ultimately develops into a steady storm displaying the classic structure of a supercell, whereas the left mover does not and continues splitting. We therefore limit all analysis of the supercell to the right-mover only, excluding the left-mover and any further storms it generates (Grant and van den Heever,

2015; Seigel and van den Heever, 2012). We define the right mover spatially, excluding the portion of the domain above a diagonal line running from (x=0km, y=78km) to (x=424.5km, y=162.5km), where x and y are the zonal and meridional coordinates respectively. This cutoff was chosen to cleanly isolate the right mover based on an examination of its precipitation, hydrometeor loading, and winds. It is important to note, however, that tracers are still produced in the left-mover region. While we do not include this emitted tracer in any of the quantities in the results section that are functions of

the total amount of tracer emitted, it is possible that tracer produced in the left-mover region could be entrained into the right-mover. This does not appear to happen to any significant degree based on our analysis of the spatial distributions of tracer and the winds over time (not shown), but we note it as a potential source of bias, as we cannot identify the gridpoint in which tracer originated.

**3 Results**

Throughout this study we will use the symbols U, V, and W to refer to the zonal, meridional, and vertical components of the wind, respectively.



## 3.1 Storm evolution and analysis

### 3.1.1 Isolated convection

The deep isolated convective cell's development follows that of a classic convective cell (e.g. Byers and Braham 1949; Wilhelmson 1974; Schlesinger 1978). The initial warm bubble forcing has developed into an updraft stretching from near the surface to the upper troposphere by t=65 minutes (Figure 3 (a, d)). Convergence at the base of the updraft is evident in the wind field. The downdrafts on either side of the main updraft are typical of isolated storm development and indicate the presence of hydrometeors, which create these downdrafts primarily by evaporative cooling and drag. Precipitation on the downshear side of the storm is favored in this weakly sheared system, and 20 minutes later at t=85 minutes the downdraft on the downshear side stretches from the upper troposphere to the surface (Figure 3 (b, e)). By this time, a surface cold pool has also developed from the evaporative cooling of the hydrometeors falling within the downdraft. The cold, dense air of the cold pool spreads laterally and completely cuts off the low-level inflow to the updraft by t=135 minutes (Figure 3 (c, f)). As a result, the updraft decays completely shortly after this.





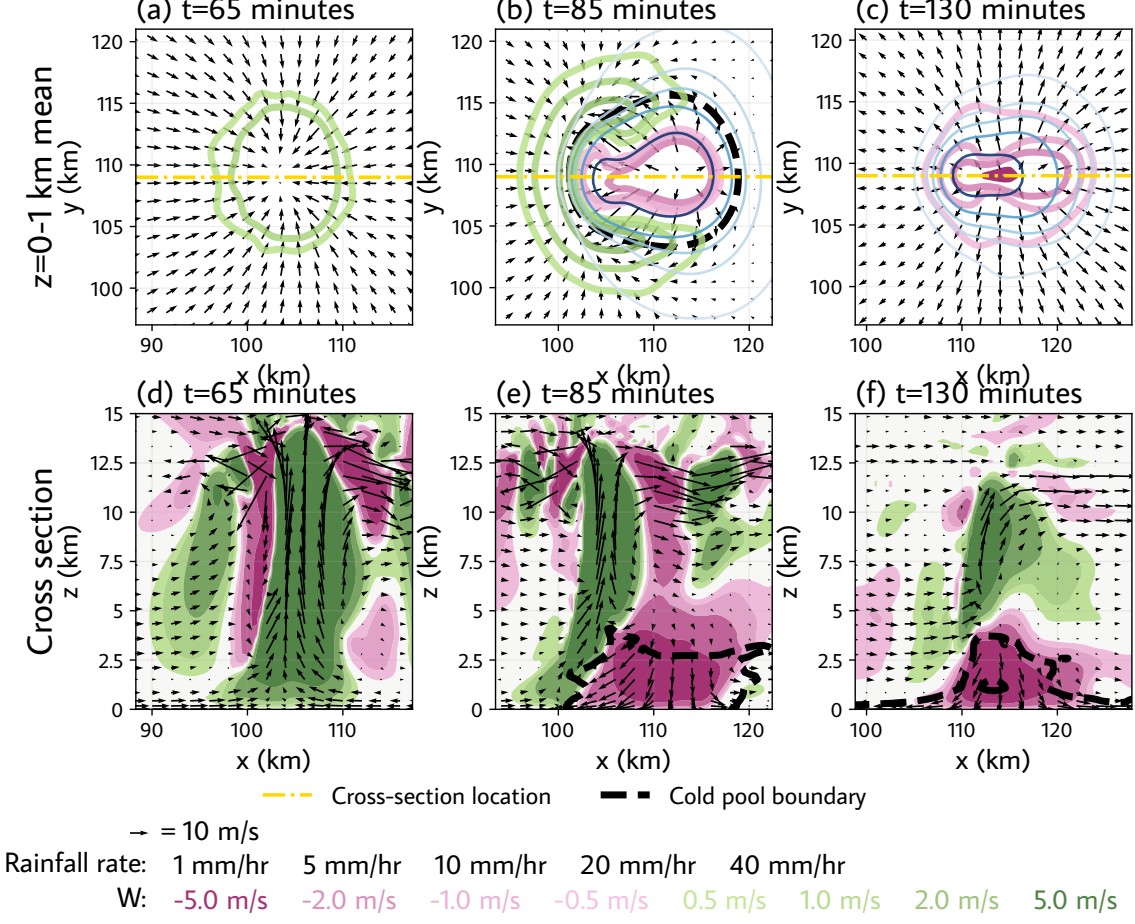

**Figure 3 Development of the isolated convective storm. Panels (a-c) show a plan view of the low-level storm features. The rainfall rate contours shown correspond to the five rainfall rate thresholds for emission of each of the rain-sourced tracer species, as described in Sect. 2.2. Panels (d-f) show cross sections through the locations indicated in panels (a-c). In all panels, the cold pool boundary is defined as the -0.04 m$^2$s$^{-1}$ buoyancy contour, following Seigel and van den Heever (2012) and Tompkins et al. (2001).**

### 3.1.2 Squall line

The squall line is initially nearly completely uniform in the line-parallel (i.e. meridional) direction (Fig. 4 (a)) due to the uniform nature of its initialization. By t=70 minutes, it has formed a cold pool with the typical cleft-and-lobe structure of a density current (Simpson, 1969). Convergence drives the updraft at the gust front, and hydrometeors are present in this updraft. At t=105 minutes, the leading line of updrafts has developed an alternating cell structure in the meridional direction (Fig. 4 (b)). Regions of strong surface downdrafts and heavy rainfall stretching ~15km behind the gust front alternate with regions of weaker downdrafts and heavy rain that only extend ~10km behind the gust front (Fig. 4 (b)). Weak ascending upper-level rear outflow is present (Fig. 4 (e, h)) (Weisman et al., 1988). The updraft attains an upshear tilt by t=155 minutes (Fig. 4 (f, i)). The updraft is continuous from the gust front to upper levels within the lobe (Fig. 4 (i)), but within the cleft it is nearly broken by a downdraft bearing hydrometeors (e.g. Jorgensen et al., 1997) (Fig. 4 (f)). The rear inflow jet has also





strengthened by this point in both the clefts and lobes (Fig. 4 (f, i)), which is in keeping with the hypothesis that it is driven
by a pressure deficit between the cold pool and updraft when the updraft tilts upshear (Weisman, 1992).

**Figure 4 Development of the squall line. Panels (a-c) are as in Figure 3 (a-c). Panels (d-f) show a cross section through one of the cleft regions that develop along the line, and panels (g-i) show a cross section through a lobe. The y values at which these cross**





**sections are taken are indicated in panels (a-c). The cold pool boundary is defined as in Figure 3. All winds shown are storm-**
**relative.**

### 3.1.3 Supercell

The storm that forms from the supercell's initial warm bubble forcing develops a mid-level cyclonic/anti-cyclonic vorticity couplet (Schlesinger, 1978) at around t=60 minutes and subsequently splits into left- and right-movers (Fujita and Grandoso,

1968; Weisman and Klemp, 1982) at about t=75 minutes (not shown). The right mover is favored due to the veering wind profile used in the environmental base state (Weisman and Klemp, 1984). At t=90 minutes, the right mover has the classic comma/hook shape (Weisman and Klemp, 1984), distinct rear- and forward-flank downdrafts with precipitation generally centered around the latter (Lemon and Doswell, 1979; Rotunno and Klemp, 1985) (Figure 5 (a)), and positive mid-level vertical vorticity indicative of the mid-level mesocyclone (Davies-Jones, 2015) (Figure 5 (d)). By t=140 minutes the storm

has acquired low-level vertical vorticity (Rotunno et al., 2017) (Figure 5 (e)), and the rain centered on the forward-flank downdraft is heavy and covers a large area (Fig. 5 (b)). Shortly after this, the low-level vertical vorticity is greater than the mid-level vertical vorticity (Figure 5 (f)), and the storm is fully mature.








Figure 5 Development of the supercell. Panels (a-c) are as in Figure 3 (a-c). These panels also indicate the portion of the domain
that contains the left-mover and which was excluded from analysis. Panels (d-f) show the positive vertical vorticity near the
updraft, at both mid- and low-levels. The low-level vorticity is the mean from 0 km – 1 km AGL, and the mid-level vorticity is the
value at 5 km AGL. Panels (g-i) show a cross section through the region indicated in panels (a-c), as in Figure 3 (d-f). The cold pool
is shown in panels (a-c) and (g-i) as in Figure 3.

### 3.2 Tracer analysis

All tracer concentrations or mixing ratios given in the remainder of this work represent the spatial mean over the updraft of each storm. Besides where it is included for emphasis, we exclude "mean in-updraft" from references to these quantities for brevity. Model gridpoints are considered in-updraft if they have a vertical velocity of at least 1 m/s and a condensate loading of at least 0.1 g/kg.

We use the term "fixed-source tracer" to refer to the *sum of all* fixed-source tracer species. Any discussion of

individual tracer species specifies explicitly that that is what is being discussed. We also use the shorthand "1 mm hr⁻¹ tracer" to refer to the rain-sourced (RS) tracer that is emitted in regions of at least 1 mm hr⁻¹ of rainfall, and likewise for the rain-sourced tracers emitted at the other rainfall thresholds.

### 3.2.1 Outline of metrics

We first outline several metrics that will be used throughout the discussion of the tracer analysis. The fixed-source tracer

mixing ratio in a gridpoint is the product of the mass of air in the gridpoint that was in contact with the surface and the duration for which it was in contact with the surface, divided by the mass of air in the gridpoint, i.e., in a single gridpoint:

$$Mean\ FS\ tracer\ mixing\ ratio\ [\#\ kg^{-1}]$$
$$= \frac{Total\ (mass * time)\ of\ surface\ contact\ [kg*s] * Emission\ rate\ [\#\ kg^{-1}s^{-1}]}{Mass\ of\ air\ in\ gridpoint\ [kg]} \quad (1)$$

The rain-sourced tracer mixing ratio can be expressed analogously, with the term for kg*seconds of surface contact being

limited to gridpoints in which it is raining at or above one of the given rainfall rate thresholds. The mixing ratio of the rain-sourced tracers is the most direct measure of the proportion of air in a storm that we would expect to contain biological particles aerosolized by rain. This quantity reflects a combination of many different processes and characteristics of each storm, and some of the most salient questions we could ask to disentangle these factors are as follows:

- How much air from the surface does the storm entrain?

- What proportion of the storm's entrained surface air is from rainy regions?

- What proportion of existing rainy surface air does the storm entrain?

- How much rain of a given intensity does the storm produce relative to the mass of its updraft?

To address these questions, we first define the mean rain-sourced tracer mixing ratio (over the whole storm rather than a single gridpoint) mathematically:

$$Mean\ RS\ tracer\ mixing\ ratio\ [\#\ kg^{-1}] = \frac{Total\ RS\ tracer\ entrained\ [\#]}{Updraft\ mass\ [kg]} \quad (2)$$

We can rewrite this in two other ways which are algebraically equivalent, and which we can map onto the physical questions posed above:

$$Mean\ RS\ tracer\ mixing\ ratio\ [\#\ kg^{-1}] = \frac{Total\ FS\ tracer\ entrained[\#]}{Updraft\ mass\ [kg]} * \frac{Total\ RS\ tracer\ entrained[\#]}{Total\ FS\ tracer\ entrained[\#]} \quad (3)$$



and

$$Mean\ RS\ tracer\ mixing\ ratio[\#\ kg^{-1}] = \frac{Total\ RS\ tracer\ entrained\ [\#]}{Total\ RS\ tracer\ emitted\ [\#]} * \frac{Total\ RS\ tracer\ emitted\ [\#]}{Updraft\ mass\ [kg]} \quad (4)$$

We summarize these measures and the physical questions they answer, as well as giving them names which we will use in the remainder of this study for convenience, in Table 2.

By analyzing the quantities described above we can determine if a storm attains greater/lesser values of the rain-sourced tracer mixing ratio because it does/does not entrain much air from the surface; it does/does not produce much rain, or much rain of a high intensity; the air from the rainy regions it does produce is/is not entrained into the storm's updraft; and/or it produces more/less rain than the other storms on a per-updraft mass basis. We first examine the rain-sourced tracer mixing ratios themselves and then analyze these constituent quantities.

| Term | Physical question answered | Name |
|---|---|---|
| $\frac{Total\ FS\ tracer\ entrained[\#]}{Updraft\ mass\ [kg]}$ | How much air from the surface does the storm entrain per updraft mass? | [Mean in-updraft] fixed-source mixing ratio |
| $\frac{Total\ RS\ tracer\ entrained[\#]}{Total\ FS\ tracer\ entrained[\#]}$ | What proportion of the storm's entrained surface air is from rainy regions? | Rain-sourced tracer fraction |
| $\frac{Total\ RS\ tracer\ entrained\ [\#]}{Total\ RS\ tracer\ emitted\ [\#]}$ | What proportion of all rainy surface air does the storm entrain? | Rain-sourced entrainment efficiency |
| $\frac{Total\ RS\ tracer\ emitted\ [\#]}{Updraft\ mass\ [kg]}$ | How much rain of a given intensity does the storm produce per updraft mass? | Rain production efficiency |

**Table 2 Definition and description of physical quantities influencing the rain-sourced tracer mixing ratio.**


### 3.2.2 Rain-sourced tracer mixing ratio

The rain-sourced tracer mixing ratio conveys information about the mass-time (i.e. kilograms*seconds) that air in each storm's updraft has spent in rainy surface regions. The maximum values of this quantity across the three storm morphologies are of similar magnitudes for the 1 mm hr⁻¹ tracer, varying by at most a factor of ~2 (Figure 6 (j-l)). As the rainfall rate





threshold increases, the organized storms' maximum values increase relative to those of the isolated convective storm (except the 40 mm hr$^{-1}$ tracer in the supercell, the reasons for which are discussed in Sect. 3.2.4). This indicates that the isolated storm and the two organized storms entrain surface air from regions of light rain in similar proportions, at least in the sense of their maxima, but that the isolated convective storm entrains less surface air from regions of heavy rain than do the two organized storm morphologies. This could occur either because the isolated storm produces relatively little heavy

rainfall, and/or because it entrains less surface air from regions of heavy rainfall. The following sections address this question and related questions for all three storms.

     The timing of the maximum values of the rain-sourced tracer mixing ratios also differs between the isolated storm and the organized storms. The isolated convective storm reaches its maximum mixing ratio of all rain-sourced tracer species at t=90-95 minutes, while all of the rain-sourced tracer species in the squall line and supercell reach their maximum values

later in the storms' lifetimes (between t=145-180 minutes). The isolated storm's tracer mixing ratios begin decreasing across all rain thresholds shortly after reaching their maximum values because the isolated convective storm decays, while the organized storms do not (Figure 7 (a)). As a result, the mixing ratios of all rain-sourced tracers in the organized storms show a generally upward trend through the durations of the simulations.

     The squall line and supercell achieve similar maximum and end-of-simulation mixing ratios to each other for the 1,

5, and 10 mm hr$^{-1}$ tracers (Fig 7 (k-l)). However, the squall line and supercell exhibit different mixing ratios of the tracers emitted in regions of heavier rain, with the squall line reaching greater maximum values of both the 20 and 40 mm hr$^{-1}$ tracers by factors of ~1.9 and ~20.2, respectively. This indicates that, if bioaerosols are aerosolized by lighter rainfall rates, the per-updraft entrainment of air containing these particles is similar between these two storm morphologies, but that the squall line entrains more of this air on a per-updraft basis if the aerosolization requires heavier rainfall.






**Figure 6 In-updraft tracer vs time, by tracer emission mechanism and species, and by storm morphology.** Panels (a-c) shows the mean in-updraft concentrations of each fixed-source tracer species (i.e. of tracers emitted during different periods of time) as a fraction of the total amount of that species that has been emitted up to that time. Panels (d-f) show the same for each of the five rain-sourced tracer species. Panels (g-i) show the same as panels (a-c) but as mean number concentrations, rather than as a
fraction of the total emitted. Panels (j-l) shows the same as (g-i) but for the rain-sourced tracer species.





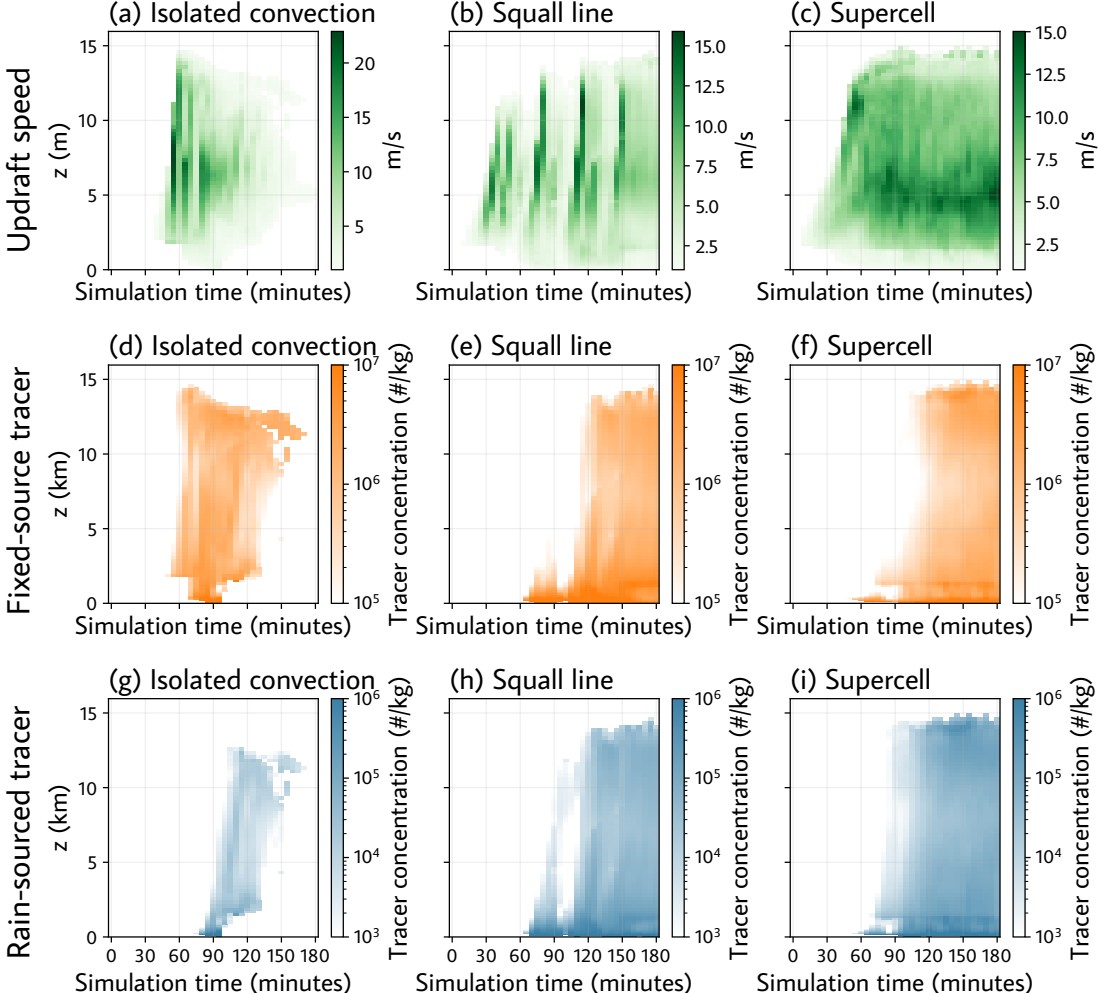

**Figure 7 Mean in-updraft vertical and temporal distributions of updraft speed and tracer concentrations, by storm. Panels (a-c) shows mean updraft speed of in-updraft points at each vertical level and time. Panels (d-f) depict the same for sum of the mixing ratios of all fixed-source tracer species. Panels (g-i) show the same for the rain-sourced tracer corresponding to the lowest rain threshold for emission (1 mm hr⁻¹).**

### 3.2.3 Fixed-source tracer mixing ratio

The fixed-source tracer mixing ratios convey information about each storm's entrainment of all surface-based air. The isolated convective storm and the squall line's maximum values of the fixed-source tracer mixing ratios are within ~5% of each other, while the supercell's maximum mixing ratio is ~30% less than that of either the isolated convection or the supercell (Figure 8 (a-c)). However, the isolated convective storm reaches this maximum mixing ratio earlier in the simulation than does the squall line, and so less fixed-source tracer has been emitted into the domain when it reaches this maximum. This implies that the isolated convective storm has the greatest peak proportion of air in its updraft that originated at the surface, as the fixed-source tracer mixing ratio indicates the mass-seconds for which the air in a gridpoint was in



contact with the surface. This generally agrees with conceptual models of each storm, where the updraft of an isolated
convective cell may be fed by a single region of convergence at the surface (Byers and Braham, 1949), while conceptual
models of squall lines often contain a region of descending rear inflow originating at mid-levels (Smull and Houze, 1987),
and the updrafts of supercells entrain some air from their own downdrafts originating at mid-levels (Rotunno and Klemp,
1985).

The supercell's low fixed-source tracer concentrations in the first ~120 minutes of the simulation owe to the
progression of the storm's development. The maximum 5km updraft speed has already reached its maximum strength of
nearly 40 m/s by t=100 minutes (Figure 9 (c)), but its low-level updraft at this time is weak (Figure 7 (c)). At t=100 minutes,
the storm has developed a midlevel mesocyclone, but the low-level mesocyclone and updraft are still developing (Figure 5
(e), Fig 7 (c)), which follows the typical progression of supercell development (e.g. Davies-Jones, 2015).





**Figure 8 Factors contributing to the mean in-updraft rain-sourced tracer mixing ratio as a function of time. Panels (a-c) show both the mean in-updraft fixed-source tracer mixing ratio and the total mass of the updraft. Panels (d-f) depict the rain-sourced tracer fraction, which is the ratio of the total in-updraft rain-sourced tracer to the total in-updraft fixed-source tracer, for each of the five rain-sourced tracers. Panels (g-i) show the rain production efficiency, which is the ratio of the total emitted rain-sourced tracer to the mass of the updraft for each of the five rain-sourced tracers. Panels (j-l) show the entrained fraction of each of the five rain-sourced tracers, which is the ratio of the total in-updraft tracer to the total amount of that tracer emitted for each tracer.**





The fixed-source tracer mixing ratio in the isolated convective storm is dominated by the tracer species that are emitted early in the simulation (Figure 6 (a, g)). The fixed-source tracer mixing ratios in the squall line and supercell updrafts are instead driven by the entrainment of newly emitted tracer as the storm progresses and encounters new surface air (Figure 6 (b-c, h-i)). This is also the reason that, for the squall line and supercell more so than for the isolated convective storm, the mixing ratios of each individual tracer species are nearly identical beginning shortly after each species is emitted: new air from the surface is being entrained by the storm, which contains all of the tracer species that have been emitted up to that point. The isolated convective storm does not entrain as much air that is in contact with the surface at later times in the simulation because its updraft decays.

### 3.2.3 Rain-sourced tracer fraction

The rain-sourced tracer fraction refers to the ratio of a storm's rain-sourced tracer mixing ratio to its fixed-source tracer mixing ratio. Physically, this describes the proportion of all entrained surface air that has been in contact with rainy regions. This quantity is calculated individually for each rain-sourced tracer species (which correspond to different rainfall thresholds).

The isolated convective storm achieves lower maximum values of the rain-sourced tracer fraction than either of the two more organized storm morphologies for all rain-sourced tracer thresholds (Figure 8 (d-f)). This provides one aspect of the explanation for the lower rain-sourced tracer mixing ratios reached in the isolated convective storm: compared to the other two storms, the surface air it entrains from rainy regions is diluted by a larger proportion of surface air from non-rainy regions. In conjunction with the fixed-source tracer mixing ratios, this indicates that the isolated convective storm's rain-sourced tracer mixing ratios reach the values they do because a larger proportion of its updraft consists of surface air, rather than because the storm is efficient at entraining air from rainy regions specifically.

Whereas the squall line and supercell both have larger maximum rain-sourced tracer fractions than the isolated convective storm across all rain-sourced tracer species, the differences in rain-sourced tracer fraction between the squall line and the supercell themselves vary by rainfall rate (Figure 8 (e-f)). The squall line's maximum and time-averaged tracer fractions are ~15-50% less than that of the supercell for the rain-sourced tracers emitted at 10 mm hr$^{-1}$ of rainfall or less, but the squall line's tracer fractions are larger for the tracers emitted at more than 20 mm hr$^{-1}$ of rainfall. In other words, the supercell's proportion of entrained surface air from regions of light rain is larger than that of the squall line, but the squall line's is larger for heavier rainfall rates.

In conjunction with the rain-sourced tracer mixing ratio and the fixed-source tracer mixing ratio, the rain-sourced tracer fraction allows us to describe the physical processes responsible for the differences in the entrainment of rainy air between the three storm morphologies. The squall line and supercell reach comparable mixing ratios of the three rain-sourced tracers corresponding to the rainfall rates less than 10 mm hr$^{-1}$ (Figure 6 (k-l)). However, we can see from the relative values of the fixed-source tracer mixing ratio (squall line > supercell) (Figure 8 (b-c)) and rain-sourced tracer fraction (supercell > squall line) (Figure 8 (e-f)) that these comparable values are reached through different transport





dynamics. The squall line entrains more air that has been in contact with the surface (in a mass*seconds sense), but less of
this air is from rainy regions. The supercell does not entrain as much air that has been in contact with the surface, but more
of what it does entrain is from regions where it is or has been raining. By contrast, for the tracers emitted at heavier rainfall
rates, the squall line reaches greater mixing ratios than the supercell (Figure 6 (k-l)). This is attributable both to the squall
line's larger amount of entrained surface air (Figure 8 (b-c)) and the fact that a larger fraction of this surface air originates in
regions of heavy rain (Figure 8 (e-f)). The isolated convective storm entrains a quantity of air from the surface that, at its
peak, is comparable to the largest proportion reached by either of the more organized storms (Figure 8 (a-c)). However, the
isolated convective storm only briefly achieves rain-sourced tracer concentrations of the same order of magnitude as the
other storms for even light rain (Figure 6 (j-l)). This is because a much smaller fraction of its entrained surface air has been
in contact with rainy regions, as evidenced in its smaller rain-sourced tracer fractions (Figure 8 (d-f)). For heavy rain, its
rain-sourced tracer fractions are even smaller, and so are the corresponding concentrations of these tracers in its updraft.

### 3.2.4 Rain production efficiency


We now analyze the quantities in the latter two rows of Table 2, the rain production efficiency and the rain-sourced tracer
entrainment efficiency, beginning with the former. The rain production efficiency measures the rain produced by the storm
relative to the mass of its updraft. It is calculated independently for each rain-sourced tracer species, as they require different
rainfall rates for emission.

405        The isolated convective storm achieves the greatest instantaneous value of the rain production efficiency for all rain
thresholds (Figure 8 (g-i)). As the rain production efficiency is the ratio of the instantaneous rainfall rate to the instantaneous
updraft mass, these large values are driven by the fact that the storm continues to rain after its updraft begins decaying
(Figure 9 (a, d)), rather than it producing an exceptionally large amount of rain per updraft mass. A more salient measure
than the maximum value of the isolated convective storm's rain production efficiency is the value at t=100 minutes, when
the storm's updraft mass (Figure 8 (a)) and the size of its rainy region (Figure 9 (d)) are both near their peak. As this
coincides with the isolated convective storm having reached a mature stage in its development, we compare the values of the
isolated storm's rain production efficiency at t=100 minutes against the end-of-simulation values of the other storms, when
they exhibit characteristics indicating their maturity as discussed in Sect. 3.1. Using this measure, the isolated convective
storm's rain production efficiency ranges from half of the squall line's for the lightest rain to a quarter of the squall line's for
the heaviest rain, and from ~half to ~80% when compared against the rain production efficiencies of the supercell in the
same way (Figure 8 (g-i)). This indicates that the squall line and supercell produce more rain per updraft mass than does the
isolated convective storm for all rain intensities, but that this difference is most pronounced for heavy rain (excepting the 40
mm hr$^{-1}$ rainfall rate in the supercell).





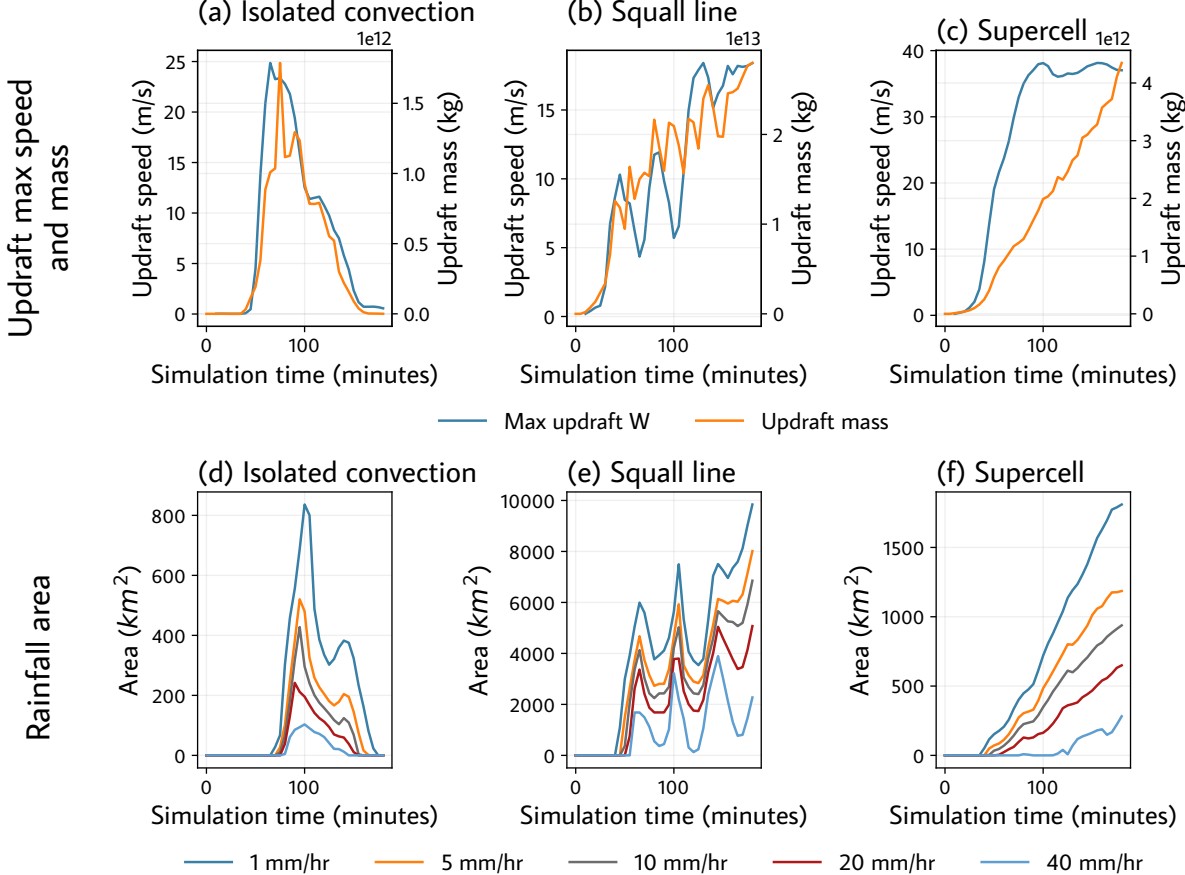

**Figure 9 Storm-averaged measures of development for each storm. Panels (a-c) show the total updraft mass and the maximum updraft speed in blue for each storm across time. The maximum updraft speed is calculated at each point in time as the largest vertical velocity met or exceed in a region of 2 km x 2 km horizontal extent at 5 km AGL, smoothed over 15 minutes. Panels (d-f) show the total area of the domain in each simulation where the rainfall rate meets or exceeds each of the five rainfall thresholds for rain-sourced tracer emission, vs time.**

At the end of the simulation when both organized storms are mature, the squall line and the supercell's rain production efficiencies for rainfall rates <= 10 mm hr$^{-1}$ are within ~25% of each other's (Fig. 8 (h-i)). For the 20 mm hr$^{-1}$ and 40 mm hr$^{-1}$ rainfall rate thresholds, the squall line attains greater time-averaged, maximum, and end-of-simulation values than the supercell (Fig. 8 (h-i)). This is especially pronounced for the 40 mm hr$^{-1}$ rain production efficiency. Supercells are known to be low precipitation efficiency storms (not to be confused with the "rain production efficiency" defined here) in that they convert a lesser portion of the vapor entrained into the storm into precipitation that reaches the ground compared to other storm morphologies because of a high degree of evaporation in their downdrafts (Browning, 1977; Foote and Fankhauser, 1973; Newton, 1966). It is therefore not surprising that, despite the squall line and supercell being initialized in identical moisture environments (Table 1) and the supercell's updraft being faster (Figure 7 (b-c), Figure 9 (b-c)) which we





might crudely approximate as being proportional to the moisture influx, the supercell still produces less rain per unit updraft
mass than does the squall line.

      These results tell us directly that the squall line simply produces more heavy rain per updraft mass than either the supercell or the isolated convective storm. Thus, without even considering factors directly related to inflow or entrainment, the squall line produces larger regions of heavy rain and thus has more surface air from these regions that it can entrain.

### 3.2.5 Rain-sourced tracer entrainment efficiency

The rain-sourced tracer entrainment efficiency refers to the amount of rain-sourced tracer present in the updraft of a storm instantaneously at a given point in time, as a proportion of the total rain-sourced tracer that has been emitted up to that point in time. This metric captures several factors about each storm, the interactions of which would be difficult to analyze (e.g. the location of its inflow, the speed of its updraft, the position of its rain relative to its updraft), but which together determine its overall ability to entrain the rainy air it has produced, and thereby to entrain air carrying rain-aerosolized bioaerosols.

The isolated convective storm reaches its peak value of rain-sourced entrainment efficiency for all rain thresholds at t=85-95 minutes (Figure 8 (j)), when its updraft mass is near its peak (Figure 8 (a)). Like some of the other extreme values reached by the isolated convective storm for the metrics being discussed, these large values are attributable to the timing of the storm's rainfall relative to its updraft development. The storm's total rainfall rate peaks at t=95 minutes (Figure 9 (d)), and the low-level updraft decays to near zero rapidly from t=95-105 minutes (Figure 7 (a)). While the storm's total updraft
mass is still relatively near its peak value at this time, essentially none of the rain-sourced tracer being produced from this time onward is entrained because there is no updraft at the surface to lift it. Some of the variation in the isolated convective storm's entrainment efficiency between the rain-sourced tracers evident in Figure 8 (j) is also explained by the fact that the average surface potential temperature in regions of 40 mm hr$^{-1}$ rain is ~2.4 K less than that of regions of 1 mm hr$^{-1}$ rain at t=90 minutes (not shown). The air from regions of heavy rain is therefore more negatively buoyant and difficult to loft.

Both the squall line and the supercell have significant fluctuations over time in their entrainment efficiencies across the rain-sourced tracers (Figure 8 (k-l)), owing to the complicated interplay of the factors contributing to this quantity. Before t=90 minutes, the squall line entrains a greater fraction of all rain-sourced tracers than does the supercell. This results physically from the later development of the supercell's low-level mesocyclone and associated low-level updraft, as discussed in Sect 3.1.3. From about t=120 minutes onward, when the supercell's low-level updraft has developed (Figure 7
(c)), the supercell entrains rain-sourced tracer from regions of light rain more efficiently than does the squall line, while the squall line entrains tracer from regions of heavy rain more efficiently than does the supercell (Figure 8 (k-l)).

      We can use the two preceding metrics to explain the trends in the rain-sourced tracer mixing ratios in a second way. The comparable values between the squall line and supercell of the mixing ratios for the tracers produced in regions with <= 10 mm hr$^{-1}$ rainfall rates (Fig. 6 (k, l)) arise because the squall line produces more light rain per unit updraft mass (Fig. 8 (h-
i)) but entrains proportionally less of the air from these regions of light rain (Fig. 8 (k-l)). These differences are both on the order of ~25%, and so the resulting rain-sourced tracer mixing ratios are of the same order of magnitude. The squall line's





larger mixing ratios for the 20 mm hr$^{-1}$ and 40 mm hr$^{-1}$ rain-sourced tracers arise because it produces more rain of this intensity per updraft mass and entrains more surface air from these regions. For the 40 mm hr$^{-1}$ tracer, the larger difference in mixing ratio from that of the supercell (Fig. 6 (k-l)) is driven by the squall line producing ~7.5x the rain of this intensity per

updraft mass that the supercell produces (Fig. 8 (h-i)). This difference is also attributable to the positioning of the storms' regions of heavy rainfall: the squall line's heaviest rain is positioned close behind the gust front, which is favorable for its entrainment (Fig. 4 (a-c)). On the other hand, the supercell's heaviest rainfall is tens of kilometers away from the surface updraft (Fig. 5 (a-c)), and so less of this air is recirculated into the updraft. The supercell's regions of light rain are also positioned nearer to its updraft, and so more air from these regions is entrained.

The maximum values of the rain-sourced tracer mixing ratios attained at about t=90 minutes in the isolated convective storm (Figure 6 (j)) arise because that is the only time in its life during which it both produces amounts of rain per updraft mass (Figure 8 (g-i)) and entrains proportions of air from these rainy regions (Figure 8 (j-l)) that are comparable to those of the other two storms. Physically, in the isolated convection there is only a brief period between the onset of rain and the development of the cold pool, which cuts off low-level inflow to the updraft. The isolated storm continues to rain

after this time as the updraft decays (Figure 9 (a, d)), but the updraft at the surface is soon no longer capable of lofting air from the rainy surface (Figure 7 (a)). Thus, the rain-sourced tracer entrainment efficiency plummets, along with the rain-sourced tracer mixing ratio.

**4 Conclusions**

We used a high-resolution cloud-resolving model to conduct idealized simulations of three archetypal midlatitude

storm morphologies: an isolated convective storm, a squall line, and a supercell, utilizing tracer quantities to identify the origins of air entrained into the storms' updrafts and their relative entrainment efficiencies. We assess the potential for each storm to entrain air containing rain-aerosolized bioaerosols by analyzing the mean concentrations of tracers emitted in rainy and non-rainy surface regions across the updraft of each storm. Our first conclusion from this analysis is that the isolated convective storm briefly entrains air from regions of light (>1 mm hr$^{-1}$) rainfall in quantities relative to the size of its updraft

that are of the same order of magnitude as those of the two more organized storm morphologies. This maximum concentration of air entrained from rainy regions is short-lived because the isolated convective storm's updraft begins decaying shortly after this maximum is reached. It reaches this maximum concentration because it entrains more surface air than either of the other two storms, but less of this surface air is from rainy regions; these factors balance to reach a comparable proportion of its updraft that is from regions of light rain compared to the other two storms. The isolated

convective storm produces less heavy rain than the other two storm morphologies, and it entrains less of the air from the regions of heavy rain that it does produce.

Our second conclusion from this analysis is that the squall line and supercell entrain comparable concentrations of air from regions with rainfall rates <= 10 mm hr$^{-1}$, but that they do so for different reasons. The squall line entrains a larger



amount of total surface air than the supercell, but a larger proportion of the surface air that the supercell entrains has been in
contact with surface rainy regions of this rainfall intensity. These differences offset each other such that the time-averaged
mixing ratios of these tracers are comparable between the two storms.

Our third conclusion is that the squall line entrains more air from regions of 20-40 mm hr$^{-1}$ rain into its updraft than
does the supercell, and that this difference (in a ratio sense) increases monotonically with rainfall rate. This arises because
the squall line produces more heavy rainfall per updraft mass than does the supercell, and it also entrains more of the air
from the regions of heavy rainfall it produces. Relative to the supercell, the squall line's values of both the rain produced per
updraft mass and the fraction of this rainy air that is entrained into the storm also increase monotonically with rainfall rate.

These results indicate that the potential for bioaerosol transport by each storm depends heavily on the rainfall rate
required for rain-induced aerosolization of these particles. For light rainfall rates, the differences between the three storm
morphologies are less pronounced. Although the isolated convective storm has only a brief time where it can entrain and
transport air from even lightly rainy regions, the amount of air from these regions in its updraft is of the same order of
magnitude as that of the organized storms. As such, the entrainment of bioaerosols aerosolized by rain into isolated
convective storms is time-limited. Any bioaerosols it does entrain in this way would not be microphysically impactful in the
developing stage of the storm because the updraft is already in the process of decaying, but it is still possible that the
introduction of warm-temperature INPs could impact the microphysical processes in the storm as it decays. The squall line
and supercell entrain similar mixing ratios of this lightly rainy surface air in their updrafts. If rain-induced aerosolization of
biological particles requires heavy rain, the isolated convective storm is less likely to be either microphysically influenced by
or an effective transporter of these aerosolized particles because it entrains very little air from regions of heavy rain. Both the
supercell and squall line entrain more air bearing these particles than the isolated convective storm, but the squall line attains
significantly greater concentrations than the supercell. Based on this finding, the squall line's potential for the transport of air
containing rain-aerosolized bioaerosols is the greatest of the three storm morphologies if this aerosolization requires heavy
rain. However, a very small number of warm-temperature INPs (such as bioaerosols) are thought to be able to glaciate
clouds through the initiation of secondary ice production (e.g. Beard, 1992; Huang et al., 2017; Mignani et al., 2019), and
thus even in this case bioaerosol-induced microphysical effects on the isolated convective storm and squall line could still be
possible.

One limitation of this study is that, by using tracers to track the locations of air carrying rain-aerosolized biological
particles, we do not account for the effects of microphysical and other aerosol-specific processes that affect bioaerosol
transport. One of the most significant of these processes in this context is wet deposition, which we noted in the introduction
has been shown to reduce the fraction of dust entrained into deep convective storms that is then able to reach the upper
troposphere (Herbener et al., 2016; Tulet et al., 2010). Throughout this study we have discussed the transport of air
containing bioaerosols, rather than of the bioaerosols themselves, because the tracers we use cannot be directly extrapolated
to bioaerosol particles for this reason.



It is important to note also that this study examines only the possibility of entrainment of a storm's rainy air by that same storm. We do not consider the possibility of the entrainment of this air by secondary convection or by a separate subsequent storm passing over the previously rainy area. While this scenario is certainly physically relevant, it is out of the scope of the current work and we leave it for future study.

While the storms we simulate are representative examples of their morphologies, it is surely possible to vary the initial conditions of each to some degree and still simulate a representative storm of each type. Chiefly, the environmental background state of each storm may influence the factors that determine the rain-sourced tracer concentrations of each storm. Changes in the strength and depth of the wind shear in which a simulated squall line is initialized have been found to impact the entrainment and precipitation characteristics of the storm (Mulholland et al., 2021), and shear has been found to correlate negatively with the precipitation efficiency (the ratio of precipitation reaching the ground to the inflowing water vapor) of convective storms in general (Foote and Fankhauser, 1973). Changes to the shear environment could therefore change the precipitation characteristics of the storms even in an identical moisture background state. While our simulations inherently represent only one possible choice of initial conditions for each storm, the initial conditions we use are well-tested and well-analyzed (Weisman and Klemp, 1982, 1984). We therefore expect that the conclusions we draw in this work generalize well to other examples of each of the three morphologies we examine here.

**Code and data availability**

All code and other materials necessary to reproduce the results presented herein are available at https://doi.org/10.5281/zenodo.15659853 (Davis and van den Heever, 2025). The RAMS output itself is not included due to file size constraints; however, the RAMS source code and namelists used for the simulations are included, and thus the RAMS output can be reproduced from the available materials. All code for producing analysis and figures from this data is included. We also include the underlying data presented in all figures, as they are of a manageable size.

**Interactive computing environment**

The file "analysis-tracerpaper.ipynb" in the repository linked in the previous section can be used to produce all figures included herein, with the caveat that the RAMS output must first be reproduced as it cannot be included due to file size constraints. This Jupyter notebook can be run by following the instructions in the repository's "README.md".



## Author contributions

CMD and SCvdH designed the simulations and analysis. CMD made developments to the RAMS source code to enable the analysis presented here. CMD conducted the RAMS simulations and performed the analysis. CMD and SCvdH prepared the paper, with contributions and edits from all co-authors.

## Competing interests

The authors declare that they have no conflict of interest.

## Acknowledgements

C. M. Davis and S. C. van den Heever acknowledge support from DOE award DE-SC0021160 and from NSF award AGS-2105938. S. Kreidenweis, R. J. Perkins, and L. D. Grant acknowledge support from NSF award AGS-2105938. C. Mignani would like to thank the Swiss National Science Foundation for its support through the Postdoc.Mobility grant (P500PN_206661). E. A. Stone acknowledges NSF award AGS-2106370.

## Financial support

C. M. Davis and S. C. van den Heever acknowledge support from DOE award DE-SC0021160 and from NSF award AGS-2105938. S. Kreidenweis, R. J. Perkins, and L. D. Grant acknowledge support from NSF award AGS-2105938. C. Mignani would like to thank the Swiss National Science Foundation for its support through the Postdoc.Mobility grant (P500PN_206661). E. A. Stone acknowledges NSF award AGS-2106370.

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
