# Peer review of "The Entrainment of Air from Rainy Surface Regions and its Implications for Bioaerosol Transport in Three Deep Convective Storm Morphologies"

_EGUsphere, 2025_

## Author Comment (AC2)

**Reviewer Comment 1**

This manuscript presents interesting research on pathways of tracers ("biological aerosol particles") in atmospheric deep convection. However, section 3.2.1 has several obscure definitions starting from equation 1. Also, some terms, physical question answered and names in table 2 contradict each other. Could you please revise this whole section. I suggest that you first explain rigorously step by step how each physical quantity discussed in the paper was calculated in the model. Second, please make sure that the names and explanations are not misleading. The rest of the paper needs some editing accordingly.

**Response:**

We thank the reviewer for their comment and suggestions. In our response, we have sought to clarify our methodology and the calculations we perform using the model output. We have amended this section to add further description and mathematical expression of the quantities that are used to compute the metrics in Table 3 (previously Table 2), including the addition of another table (Table 2) describing our calculations. We could not identify the contradictory physical questions or names in Table 3 (previously Table 2). As specific examples or further explanation of the terms found to be contradictory or misleading were not provided, we have not been able to adjust them. However, we hope that the changes that have been made address any concerns that the reviewer may have in this regard.

**Changes:**

1. **Lines 264-273 have been replaced with:**

We first define quantities in Table 2 that are calculated directly from model output and which will be used in constructing metrics that describe the characteristics of each storm.

| Quantity | Units | Method of calculation |
|---|---|---|
| **In-updraft** | N/A | $\left(Vertical\ velocity \geq 1\frac{m}{s}\right) AND$ $\left(condensate\ mixing\ ratio \geq 0.1\frac{g}{kg}\right)$ |
| **Updraft mass** | $kg$ | $\sum_{\substack{In-updraft \\ gridpoints}} Air\ density\ \left[\frac{kg}{m^3}\right]$ $* gridpoint\ volume\ [m^3]$ |
| **Total fixed-source tracer entrained** | # | $\sum_{\substack{In-updraft \\ gridpoints}} Fixed\text{-}source\ tracer\ mixing\ ratio\ \left[\frac{\#}{kg\ air}\right]$ $* air\ density\ \left[\frac{kg}{m^3}\right]$ $* gridpoint\ volume\ [m^3]$ |

[Figure]

| | | |
|---|---|---|
| **Total rain-sourced tracer entrained** | # | $\sum\limits_{\substack{In-updraft \\ gridpoints}}$ Rain-sourced tracer mixing ratio $\left[\dfrac{\#}{m^3}\right]$

 $*$ air density $\left[\dfrac{kg}{m^3}\right]$
 $*$ gridpoint volume $[m^3]$ |
| **Total fixed-source tracer emitted** | # | $\sum\limits_{Time=0}^{Present}\sum\limits_{\substack{Surface \\ gridpoints}}$ Tracer emission rate $\left[\dfrac{\frac{\#}{kg\ air}}{s}\right]$

 $*$ air density $\left[\dfrac{kg}{m^3}\right]$
 $*$ gridpoint volume $[m^3]$
 $*$ timestep $[s]$ |
| **Total rain-sourced tracer emitted** | # | $\sum\limits_{Time=0}^{Present}\sum\limits_{\substack{Surface \\ gridpoints}}$ Tracer emission rate $\left[\dfrac{\frac{\#}{kg\ air}}{s}\right]$

 $*$ air density $\left[\dfrac{kg}{m^3}\right]$
 $*$ gridpoint volume $[m^3]$
 $*$ timestep $[s]$
 $*\begin{cases}0, & Rainfall < threshold \\ 1, & Rainfall > threshold\end{cases}$ |

**Table 2 Quantities used in constructing analysis metrics, as calculated directly from model output.**

The mixing ratio of a rain-sourced tracer in a given gridpoint represents the product of the mass of air in that gridpoint that has been in contact with the surface and the duration for which this contact occurred. Each rain-sourced tracer species captures this product of mass and time of surface contact for rainfall of a different intensity. *The mixing ratios of the rain-sourced tracers are therefore the most direct measure of the proportion of air in a storm that we would expect to contain biological particles aerosolized by rain.* These quantities reflect a combination of many different processes and characteristics of each storm, and some of the most salient questions we could ask to disentangle these factors are as follows:

---

## Author Comment (AC3)

**Reviewer comment M0:**

This manuscript uses the high-resolution RAMS cloud-resolving model to compare three types of deep convection (isolated deep convection, a squall line, and a supercell) in terms of how they entrain and transport "rain-sourced near-surface air." Overall, the topic is novel, the model/diagnostic design is sound, and the results are interesting with potential implications for cloud–aerosol interactions. However, several important issues need to be addressed before the paper can be considered for publication.

**Response:**

We thank the reviewer for their thorough and constructive comments on the manuscript. In our responses below, we have worked to clarify the rationale behind our tracer methodology and to strengthen the connection between our analysis and the entrainment of bioaerosols. The reviewer's note on the importance of considering upper-level transport has proved especially helpful for improving the manuscript, and we have added further analysis to address this point.

In our response below we assign identifiers to each reviewer comment (e.g. M0) in order to reference them more easily; reviewer comments under "major issues" are prefixed with M, and comments under "detail issues" are prefixed with D. As we have added several sections and figures to the manuscript, all figure/section numbers in the changes to the manuscript reflect the new figure/section numbers after the inclusion of all changes below.

**Changes:**

See responses to individual comments below.

**Reviewer comment M1 and M2:**

Major issues

1. The "rain-sourced" tracer is purely passive and only subject to advection and diffusion. It does not include wet deposition or particle-specific processes. Although the authors briefly acknowledge this limitation, there is no quantitative estimate or sensitivity test to assess how wet scavenging might reduce the actual amount of rain-sourced aerosol that could be lofted.

2. Because the tracers ignore wet scavenging, the only real difference between "rain-sourced" and "fixed-source" tracers is whether they are released during rainfall or not. In essence, the study shows the effect of storm winds on air masses, rather than the specific role of rainfall. As such, the link to rain-induced aerosolization (and especially to bioaerosols/INPs) feels somewhat tenuous, and the extrapolation to microphysical or health impacts is weak in its current form.

**Response:**

We agree with the reviewer's assessment of the limitations of our tracer setup. This approach was selected to first identify the storm-scale characteristics of each morphology that drive its ability to entrain air from rainy regions, without yet considering the microphysical effects from actual aerosol particles and the additional complexity they bring to the analysis. Using microphysically active aerosol particles rather than tracers produces feedbacks to the development and dynamics of the storm itself. We believe that an analysis of the factors driving entrainment of the air bearing these particles is not only itself a worthwhile scientific contribution but is also necessary groundwork for future analyses of storm feedbacks resulting from the inclusion of microphysically active aerosols, rather than inert tracers, emitted by rain.

We have now added additional discussion on this point to the introduction to clarify our approach. Furthermore, we have added estimates of surface INP concentrations aerosolized by rain (see reply to this reviewer's Comment D5 below).

**Changes:**

1. **Lines 94-100 have been replaced with:**

This study seeks to extend our understanding of bioaerosol entrainment and transport dynamics in deep convective storms by analyzing the transport of air from surface regions where rain can aerosolize biological particles using high-resolution simulations. Our goal in this study is therefore to assess whether the air containing the bioaerosols that are emitted from the surface during a storm could be entrained into that storm by the storm's winds, as well as to determine the factors that control this entrainment. We analyze this entrainment in three different convective storm morphologies and measure the entrainment and transport of this air using passive tracer quantities. These tracers are subject to advection and diffusion but not to aerosol-specific processes such as wet scavenging or nucleation, and so they act as tracers for the movement of air rather than of aerosols.

This study is the first to investigate the factors that govern storms' ability to entrain bioaerosol-enriched air from their own rainy surface regions. By first identifying these dynamical factors without needing to consider the microphysical feedbacks that would be induced by the inclusion of true rain-emitted aerosols, we are able to separate out these feedbacks from the basic storm dynamics. A limitation of this approach is that it fundamentally measures the entrainment of air, rather than of aerosols. Nonetheless, understanding how this air is entrained is a necessary precursor to determining entrainment pathways of rain-emitted bioaerosols in convective storms in future work and, ultimately, to assessing their microphysical feedbacks on the storms that entrain them.

**Reviewer comment M3:**

3. The main conclusion is that, if rain-induced bioaerosols exist, storms entrain them to different degrees. However, the paper does not show how much of the rain-sourced tracer actually reaches the upper troposphere (e.g., 8–12 km). This is critical for evaluating possible impacts on ice nucleation near cloud tops or for long-range transport. Prior studies have specifically used "lofting to 10 km" as a benchmark, but the manuscript provides no such metric.

**Response:**

We thank the reviewer for this helpful comment. We have added new sections (2.4 and 3.4) containing methods and analysis as well as a new figure (Fig. 10) to address this point. We quantify and compare the storms' transport to upper levels by calculating the tracer concentrations at upper levels. The results show that the degrees of entrainment of surface air into updrafts also apply to transport of surface air to upper levels. We also find that these results are not sensitive to the choice of height threshold by which we define "upper levels", within a reasonable range of values (discussed further in the changes below).

**Changes:**

1. **Lines 194-198 have been replaced with:**

It is important to note, however, that tracers are still produced in the left-mover region. We do not include this emitted tracer in any of the results assessing the entrainment of tracers into the storms' updrafts (Section 3.3) that are functions of the total amount of tracer emitted. It is possible, however, that tracer produced in the left-mover region could be entrained into the updraft of the right-mover. This does not appear to happen to any significant degree based on our analysis of the spatial distributions of tracer and the winds over time (not shown), but we note it as a potential source of bias, as we cannot identify the grid box in which tracer originated. We do include the tracer emitted in the left-mover region in the results assessing upper-level transport of tracers (Section 3.4), the reasoning for which is discussed in Section 2.4.

**2. Inserted at line 199:**

**2.4 Calculation of upper-level tracer concentrations and mass fluxes**

We conduct an analysis of total tracer amounts at upper levels ($z \geq 10$ km AGL) to understand the transport of surface air to the upper troposphere. The total upper-level tracer amount is calculated as:

$$Total\ upper\text{-}level\ tracer\ amount\ [\#] =$$

$$\sum_{\substack{Gridpoints \\ at\ z \geq 10\ km}} \rho \left[\frac{kg\ air}{m^3}\right] \cdot \Delta x[m] \cdot \Delta y[m] \cdot \Delta z[m] \cdot tracer\ mixing\ ratio \left[\frac{\#}{kg\ air}\right] \quad (1)$$

We normalize the upper-level tracer amounts by each storm's cumulative mass flux to upper levels. When normalized by the upward mass flux to upper levels, the upper-level tracer amounts can be fairly compared between storm morphologies. This normalization also makes it so that the upper-level tracer amounts physically represent the proportion of the storm's flux to upper levels that has been in contact with the (rainy or dry) surface (on a mass*time basis, as discussed in section 3.3.1). We normalize both the fixed-source and rain-sourced tracer amounts in this way.

We calculate the mass flux to upper levels using a limited set of physical variables output at 2 second frequency so that we can assume constant air density and vertical motion between output times. We calculate the cumulative upward mass flux as:

$$F_{upward}[kg] = \sum_{Time=0}^{Present} \sum_{\substack{Gridpoints \\ at\ z=10\ km}} \rho\left[\frac{kg}{m^3}\right] \cdot w\left[\frac{m}{s}\right] \cdot \Delta x[m] \cdot \Delta y[m] \cdot \Delta t[s] \cdot \begin{cases} 0, w \leq 0\frac{m}{s} \\ 1, w > 0\frac{m}{s} \end{cases} \quad (2)$$

Additionally, we perform separate calculations in which the upper-level rain-sourced tracer amounts are normalized by the total amount of each tracer emitted. These normalized tracer amounts can then be compared directly against in-updraft tracer concentrations in Section 3.3 that are also normalized by the total amount of each tracer emitted.

Unlike the updrafts of the left- and right-movers of the supercell, which are clearly separated, there is some overlap between the regions of upper-level tracer that are lofted by the left-mover versus by the right-mover of the supercell (not shown). As we cannot cleanly separate tracer lofted by the left- versus right-mover, in Fig. 10 (c, f) we calculate the supercell's z=10 km mass flux and upper-level tracer amounts over the entire domain including the left-mover region. Similarly, in Fig. 10 (i) we normalize by the amount of tracer emitted in the whole domain. We have compared these results against the same calculations limited only to the right-mover region, and while the magnitude of some figures presented in Section 3.3.6 differ between the approaches, the conclusions we draw from the results are the same.

We also calculate all of the results presented in Section 3.3.6 with "upper levels" defined as $z \geq 7$, 8, and 9 km AGL (not shown) in addition to our actual threshold of $z \geq 10$ km to assess the sensitivity of our results to this choice of height threshold. We find no sensitivity to this threshold that would impact any of our conclusions.

**3. Inserted at line 483:**

**3.4 Surface air transport to the upper troposphere**

In addition to effects in the updrafts of the entraining storms, the transport of aerosols by convective storms to the upper troposphere can have significant impacts on anvil ice formation (Fan et al., 2010; Saleeby et al., 2016) and convective cloud top temperature (Li et al., 2017). Aerosols that are transported to the upper troposphere also have significantly longer residence times than those that remain in the lower troposphere (Srivastava et al., 2018), and so are expected to be transported longer distances. We therefore investigate the tracer amounts at upper levels ($z \geq 10$ km AGL) for each storm morphology in order to quantify each storm's transport of surface air to the upper troposphere.

[Figure]

**Figure 10 Upper-level tracer amounts as a function of time, for each storm morphology and tracer type. Panels (a-c) show the total fixed-source tracer amount [#] divided by the cumulative upward mass flux [kg] at the z=10km level. Panels (d-f) show the total rain-sourced tracer amount [#] divided by the cumulative upward mass flux [kg] at the z=10km level. Panels (g-i) show the total rain-sourced tracer amount [#] divided by the total emitted rain-sourced tracer amount [#]. Results in panels (d-i) are shown for each of the five rain-sourced tracers.**

The results for tracer amounts at upper levels broadly parallel those of the in-updraft tracer concentrations. The isolated storm briefly lofts surface air to upper levels with an efficiency (on a per-mass flux basis) comparable to the largest value reached by either of the other storms. However, the organized storms (squall line and supercell) have surpassed it by factors of up to 2.5x at the end of the simulations (Fig. 10 (a-c)). The isolated storm's transport of rainy surface air to upper levels is minimal compared to the other storms, whether normalized by its flux to upper levels (Fig. 10 (d-f)) or by the total amount of rain-sourced tracer emitted (Fig. 10 (g-i)). In general, by the end of the simulations, only a small fraction of the total emitted rain-sourced tracer reaches the upper levels of the storms (<1%, 5%, and up to 12% for the 1 mm hr$^{-1}$ tracer in the isolated convection, squall line, and supercell, respectively, Fig. 10 (g-i)). This is in keeping with the previous analysis in Section 3.3 showing that the isolated storm entrains dry surface air into its updraft effectively, but not so for rainy surface air. The squall line transports rainy air to the upper troposphere nearly uniformly across rainfall intensities (relative to the size of the area raining at a given intensity) (Fig. 10 (h)), paralleling its rain-sourced tracer entrainment efficiency (Fig. 9 (k)). The supercell transports air from lightly rainy regions to the upper troposphere in larger quantities than it does air from intensely rainy regions, both in absolute terms (Fig. 10 (f)) and as proportions of the size of its rainy regions of each intensity (Fig. 10 (i)), again paralleling its rain-sourced tracer entrainment efficiency (Fig. 9 (l)). These results suggest that supercells are more effective than squall lines at transporting air from surface regions of light rain to the upper troposphere, squall lines are more effective at transporting air from surface regions of heavy rain to the upper troposphere, and isolated convective storms are far less effective at this transport than either of the other storms at any rain intensity.

4. **Inserted at line 507:**

Our fourth conclusion is that trends in the amount of surface air that each storm transports to upper levels parallel the trends in the amount of surface air that the storms entrain into their updrafts. This suggests that the supercell (squall line) is a comparatively more effective transporter of air from surface regions of light (heavy) rain to the upper troposphere, and that the isolated convective storm is less efficient at this transport than either of the other storms.

5. **Section 3.3 has been renamed to "Surface air entrainment into updrafts"**

**Reviewer comment D1:**

1. The captions of Figures 7–9 should clearly state variable units, normalization methods, and any temporal smoothing or averaging. e.g., Fig. 9: maximum updraft at 5 km AGL smoothed over 15 min. Currently, some details appear only in the text, which makes the figures harder to interpret.

**Response:**

We have added units to the captions of figures 7-9 (now figures 8-10) and clarified our smoothing method in the caption of figure 9 (now figure 10). We also clarified the captions' existing language where appropriate.

**Changes:**

1. **The captions of figure 8-10 now read:**

**Figure 8 Mean updraft speed and mean in-updraft tracer concentrations as a function of time and height, for each storm morphology. Panels (a-c) show the updraft speed [m s$^{-1}$], panels (d-f) show the mean fixed-source tracer mixing ratio [# kg$^{-1}$], and panels (g-i) show the mean mixing ratios [# kg$^{-1}$] of the rain-sourced tracer corresponding to the lowest rain threshold for emission (1 mm hr$^{-1}$).**

**Figure 9 Factors influencing the mean in-updraft tracer mixing ratios as a function of time, for each storm morphology. Panels (a-c) show the mean in-updraft fixed-source tracer mixing ratio [#/kg] (in blue) and the total mass of the updraft [kg] (in orange). Panels (d-f) depict the rain-sourced tracer fraction, which is the ratio of the total in-updraft rain-sourced tracer amount [#] to the total in-updraft fixed-source tracer amount [#]. Panels (g-i) show the rain production efficiency, which is the ratio of the total emitted rain-sourced tracer amount [#] to the mass of the updraft [kg]. Panels (j-l) show the entrained rain-sourced tracer fraction, which is the ratio of the total in-updraft rain-sourced tracer amount [#] to the total emitted rain-sourced tracer amount [#]. Results in panels (d-i) are shown for each of the five rain-sourced tracers.**

**Figure 10 Measures of storm development as a function of time, for each storm morphology. Panels (a-c) show the total mass of the updraft [kg] (in orange) and the smoothed maximum updraft speed [m s$^{-1}$] (in blue). The smoothed maximum updraft speed is calculated at each point in time as the largest vertical velocity met or exceeded in a region of 2 km x 2 km horizontal extent at 5 km AGL. This vertical velocity is then smoothed by taking its mean over the preceding 15 minutes. Panels (d-f) show the total area of the domain [km$^2$] where the rainfall rate [mm hr$^{-1}$] meets or exceeds the threshold for rain-sourced tracer emission. for each of the five rain-sourced tracers.**

**Reviewer comment D2:**

2. Model data are saved every 5 minutes (Table 1). Since some entrainment/updraft processes may evolve on shorter timescales, please explain whether this temporal resolution is sufficient for your analysis.

**Response:**

While model output is saved every 5 minutes, the model timestep itself is 1 second. This is a timestep comparable to those commonly used in the literature for mesoscale modeling of storm-scale processes (e.g. Mulholland et al., 2021 (3.5 s); Lebo and Morrison, 2014 (2.5 s); Drager et al., 2020 (1.5 s); Leung et al., 2023 (0.75 s); Grant and van den Heever, 2018 (0.5 s)) and we are confident that the model itself resolves processes evolving on temporal scales significantly finer than 5 minutes. The question then becomes whether we miss any significant evolution in the storm dynamical processes and/or tracer distributions by looking at output at 5-minute intervals. As our analysis largely focuses on trends in updraft mass, rainfall rate/distribution, and tracer concentration, rather than transient spikes in these quantities as might be observed with more frequent model output, variations on timescales shorter than 5 minutes are unlikely to affect our results. While we do discuss the maxima of these quantities, these again occur as part of relatively smooth timeseries of these quantities, and thus only small changes in the timing or exact magnitude of these quantities should be observed with more frequent output. Such changes seem highly unlikely to impact our conclusions. Further, the analysis of upper-level tracer concentrations that we have added (see response to this Reviewer's Comment M3 above) would be expected to capture the net effects of any finer-temporal-resolution fluctuations in entrainment missed between the analyzed 5 minute model output times. As our results for tracers at upper levels parallel those for in-updraft tracers, we have further reason to be confident that we are not missing any significant evolution with this model output timestep.

**Changes:**

None

**Reviewer comment D3:**

3. The y boundary of Squall line is cyclic, while the other cases are radical. Please discuss whether this introduces differences and why different boundaries are adopted.

**Response:**

The y boundary of the squall line is chosen to be cyclic so as to emulate the structure of a long squall line in a way that is computationally manageable. The effect of the periodic meridional boundary is to make the squall line "infinite" in the meridional direction and thus to make the storm we simulate equivalent to the middle of a very long storm. Our tracer results are all normalized in a way such that the actual size of the storm, and thus this "infinite" length, does not influence the results. This approach has been used to model squall lines a number of times previously in the literature (e.g. Lebo and Morrison, 2014; Mulholland et al., 2021; Weisman et al., 1988). The other storms (and the squall line in the x direction) are positioned so that they do not reach the boundaries of the domains, and thus the boundary conditions should not exert significant influence either way. Gravity waves could, however, reach the domain edges, since their propagation speeds are generally faster than those of the

convective systems. The radiative boundary conditions are designed to allow gravity waves to exit the domain without reflection. We have added the following sentence to explain this choice further and reference previous examples:

**Changes:**

1. **Inserted at line 138:**

A periodic boundary is commonly employed in simulations of squall lines in order to facilitate the study of a long squall line with a domain size that is computationally feasible (e.g. Lebo and Morrison, 2014; Mulholland et al., 2021; Weisman et al., 1988).

**Reviewer comment D4:**

4. The assumption that aerosol release occurs once rainfall intensity passes certain thresholds should be supported by references.

**Response:**

Recent work on this topic suggests that INP enhancement by rain correlates strongly with several measures of precipitation events, most strongly with cumulative rainfall kinetic energy and total amount of rain (Mignani et al., 2025). We apply results from this study to calculate surface INP concentrations in our response to comment D5 below. However, while our rainfall thresholds for rain-sourced tracer emission cover a broad range of rainfall intensities, they are fundamentally a simplifying assumption of our approach rather than an implementation of this observed physical mechanism. We expect that the total emitted rain-sourced tracer and the surface INP concentration estimates we calculate in our response to comment D5 would be strongly correlated. However, we do not attempt to map the two onto each other, as the tracer quantities in our approach track the movement of air rather than of aerosol particles.

**Changes:**

See response and related changes to the Reviewer's comment D5

**Reviewer comment D5:**

5. Can you provide at least a rough estimate of how the tracer concentrations could map onto INP enhancements, e.g., within the DeMott (2010) framework, and what impact this might have on ice nucleation rates?

**Response:**

We have added estimates of surface concentrations of INPs aerosolized by rain in a new section of the manuscript, Section 3.2. However, to consider the transport of these INPs into microphysically active regions of the storms (and thus assess their impact on ice nucleation rates) would require treating their aerosol-specific behaviors, including processes like deposition, nucleation, and settling. Such a treatment is outside of the current scope of this work but is a topic for future study, one which we expect to build on the current results identifying entrainment and transport pathways.

We estimate concentrations of INPs aerosolized by raindrop impact at the surface by applying relationships between precipitation parameters and near-surface atmospheric INP concentrations reported by Mignani et al. (2025) based on observations of midlatitude spring-/summer-time convective rainfall. We find that the rain from the storms we consider here could lead to surface warm-temperature INP concentrations comparable to large values reported in field studies (Kanji et al., 2017). Thus, these warm-temperature INPs aerosolized by rain have the potential to impact microphysical processes in the storm even if only a small fraction of them is entrained into the parent storm.

**Changes:**

1. **Inserted at line 254:**

**3.2 Estimates of warm-temperature INP concentrations at surface level**

We estimate concentrations of warm-temperature INPs (-8°C and -10°C) emitted by rain at the surface using a parameterization provided by Mignani et al. (2025) (hereafter M25). For reasons discussed in Sections 1 and 2, we do not attempt to model the transport or in-cloud concentrations of INPs released in this way. In-cloud INP concentrations will be less than the surface concentrations we calculate here due to mechanisms including below-cloud scavenging by rain and in-cloud wet deposition, even if the emitted INPs are entrained completely. We conduct this estimate to ascertain whether the storms' rain aerosolizes large enough quantities of INPs to have the potential for microphysical influence even with depletion of these concentrations in-cloud.

The parameterization of M25 provides linear fits in log-log space between INP concentrations and cumulative precipitation amount. We apply these observed relationships to map accumulated rainfall values at the time of maximum instantaneous rainfall for each storm (t=95 minutes for the isolated storm, t=145 minutes for the squall line, and t=180 minutes for the supercell) to total released INP concentrations. The INP concentration versus rain relationships themselves are derived from observations of late springtime mid-latitude convective rainfall, which is an appropriate setting for the storm morphologies we simulate. We assume INP concentrations of 0 L$^{-1}$ at standard temperature and pressure (STP) where precipitation amounts

are below the minimum observed in M25 (0.001 mm). For precipitation amounts above the maximum observed (9.557 mm), we clip our accumulated precipitation amount to this value before calculating INP concentrations to avoid extrapolation beyond the observed range of M25. We focus our analysis on INPs with freezing temperatures of -8 °C and -10 °C because INPs that freeze at such warm temperatures are typically much less abundant than INPs that freeze at colder temperatures (DeMott et al., 2010). INPs with these warm freezing temperatures are therefore expected to experience the largest relative concentration increases upon entrainment of bioaerosols.

At surface level, the median rain-sourced concentrations of INPs active at -10 °C is 0.06 L$^{-1}$ STP, 0.33 L$^{-1}$ STP, and 0.20 L$^{-1}$ STP for the isolated convection, the squall line, and the right-mover of the supercell, respectively (Fig. 6). Maximum values reach up to 0.48 L$^{-1}$ STP for each storm morphology (Fig. 6). This is comparable to large values reported from field studies (Kanji et al., 2017). Garcia et al. (2012) demonstrated the microphysical potential of these INP concentrations. They found freezing onset temperatures as high as -6 °C from surface biological INP concentrations as small as 0.01 L$^{-1}$. Our results indicate that the rainfall produced by all three storm morphologies aerosolizes warm-temperature INPs that, if entrained, have the potential to exert microphysical influence on the storms' evolution.

[Figure]

**Figure 6 Plan views of estimated surface INP concentrations aerosolized by rain, for each storm and INP freezing temperature (-8°C and -10°C). Surface INP concentrations are estimated from simulated rainfall amounts using the M25 parameterization and are shown in units of L$^{-1}$ STP. Median INP concentrations are calculated over regions where the INP concentration > 0 L$^{-1}$ STP, and only over the right-mover of the supercell, which is shown as in Figure 3. Note that rain-**

sourced tracers are released in the model once rainfall intensity passes defined thresholds, which simplifies the physical mechanism of INP release by raindrop impact (M25).

**Reviewer comment D6:**

6. Table 1 units: The initial aerosol profile is listed with units of "mg⁻¹," which seems inconsistent. Please correct the unit and confirm the magnitude.

**Response:**

Particle number per milligram (of air) is the unit in which the surface aerosol concentration is specified in the model namelist. We have changed this in the table to read "particles per mg of air" for clarity. While (#) $cm^{-3}$ may be a more standard unit, this can be easily obtained by multiplying by air density. For reproducibility considerations with the model namelist settings, we have left the unit as $mg^{-1}$.

**Changes:**

1. **The last sentence of Table 1 has been changed to:**

Sulfates, log-normally distributed with median radius of 75 nm; concentration decreases exponentially with height, from 500 particles per mg of air at the surface with e-folding height of 7 km